# Interpretation and Mapping Tree Crown Diameter Using Spatial Heterogeneity in Relation to the Radiative Transfer Model Extracted from GF-2 Images in Planted Boreal Forest Ecosystems

Zhaohua Liu [1,2,3] , Jiangping Long [1,2,3,*] , Hui Lin [1,2,3], Kai Du [4,5], Xiaodong Xu [1,2,3], Hao Liu [1,2,3], Peisong Yang [1,2,3], Tingchen Zhang [1,2,3] and Zilin Ye [1,2,3]

1   Research Center of Forestry Remote Sensing & Information Engineering, Central South University of Forestry and Technology, Changsha 410004, China
2   Key Laboratory of Forestry Remote Sensing Based Big Data & Ecological Security for Hunan Province, Changsha 410004, China
3   Key Laboratory of State Forestry Administration on Forest Resources Management and Monitoring in Southern Area, Changsha 410004, China
4   Key Laboratory of Tibetan Plateau Land Surface Processes and Ecological Conservation (Ministry of Education), Qinghai Normal University, Xining 810008, China
5   Qinghai Province Key Laboratory of Physical Geography and Environmental Process, College of Geographical Science, Qinghai Normal University, Xining 810008, China
*   Correspondence: longjiangping@csuft.edu.cn; Tel.: +86-0731-8562-3848

**Abstract:** Tree crown diameter (CD) values, relating to the rate of material exchange between the forest and the atmosphere, can be used to evaluate forest biomass and carbon stock. To map tree CD values using meter-level optical remote sensing images, we propose a novel method that interprets the relationships between the spectral reflectance of pixels and the CD. The approach employs the spectral reflectance of pixels in the tree crown to express the diversity of inclination angles of leaves based on the radiative transfer model and the spatial heterogeneity of these pixels. Then, simulated and acquired GF-2 images are applied to verify the relationships between spatial heterogeneity and the tree CD. Meanwhile, filter-based and object-based methods are also employed to extract three types of variables (spectral features, texture features, and spatial heterogeneity). Finally, the tree CD values are mapped by four models (random forest (RF), K-nearest neighbor (K-NN), support vector machine (SVM), and multiple linear regression (MLR)), using three single types of variables and combinations of variables with different strategies. The results imply that the spatial heterogeneity of spectral reflectance is significantly positively correlated with tree CD values and is more sensitive to tree CD values than traditional spectral features and textural features. Furthermore, the ability of spatial heterogeneity to map tree CD values is significantly higher than traditional variable sets after obtaining stable features with appropriate filter window sizes. The results also demonstrate that the accuracy of mapped tree CD values is significantly improved using combined variable sets with different feature extraction methods. For example, in our experiments, the $R^2$ and rRMSE values of the optimal results ranged from 0.60 to 0.66, and from 15.76% to 16.68%, respectively. It is confirmed that spatial heterogeneity with high sensitivity can effectively map tree CD values, and the accuracy of mapping tree CD values can be greatly improved using a combination of spectral features extracted by an object-based method and spatial heterogeneity extracted by a filter-based method.

**Keywords:** tree crown diameter; GF-2; the LESS model; spatial heterogeneity; feature extraction



## 1. Introduction

The forest canopy is a basic source of forest energy through photosynthesis and respiration that achieves material exchange between forest ecosystems and the atmosphere [1].

The average crown diameter (CD) is viewed as one of the most important parameters, determining the number and spatial distribution of leaves on the crown and thus affecting the rate of material exchange between the forest and the atmosphere [2]. Accurately obtained CD values can be further used to map forest biomass and carbon stock [3–5]. Recently, due to the complex ecological conditions of forests and the spatial and temporal limitations of field investigations, remote sensing technology has become an important tool for forest mapping and dynamic monitoring [6–9]. However, the tree crown diameter (CD) remains a challenging parameter to be accurately estimated from images with common spatial resolution, due to the mixed signals received by a single pixel, which arise from the forest crown as well as the background in forested areas [2,10].

A common premise underlying CD measurement is that the spatial resolution of images is much smaller than the size of the crown [10,11]. Remote sensing images with medium or low spatial resolution (Modis, Landsat, and Sentinel-2) are inadequate for rationally extracting forest crown information. In a previous study, Li and Strahler developed a geometric optical model (the Li–Strahler model) to successfully map tree CDs using Landsat images at the stand level, but it is difficult to meet the accuracy requirements with this method [12]. Most studies focus on high-spatial-resolution images acquired from aerial photographs and optical satellite images with sub-meter levels (Quickbird, Worldview, and Geoeye) [10,13–15]. Naturally, image segmentation is widely applied to obtain the numbers and shapes of forest crown using these high-spatial-resolution images. However, the cost of acquired optical satellite images with sub-meter levels and the burdens of computation limit the application of mapping tree CD values in large areas. Therefore, optical satellite images with meter levels are regarded as a good choice for mapping tree CDs [2].

In general, determining the sizes of the tree crown requires the numbers and shape of the canopy to be obtained using various image segmentation methods. Asner et al. investigated the possibility of mapping CD values using high-resolution satellite images by manually depicting the canopy on IKONOS images [16]. Palace et al. developed an automated crown detection and depiction algorithm and found that it provided a better estimation of tree CD values compared to manual depiction [17]. Furthermore, it is also confirmed that detecting and depicting the shape of forest canopies relies on the proportional relationship between the pixel size and crowns [14,18,19]. Song et al. showed a strong correlation between the ratio of image semi-variance functions and mean CD values for different spatial resolutions [2]. Pouliot et al. investigated the effect of the spatial resolution of images on canopy depiction, and the results demonstrated that the optimal ratio of CD to pixel size was 15:1, with a smaller ratio failing to depict the canopy boundaries and a larger ratio providing too much variation in brightness within the crown [18]. Although images with a higher resolution help detect smaller trees, they generate larger intra-class variance for larger crowns, which leads to larger commissioning errors [13,20]. Essentially, accurately recognizing tree CD values with meter-level images depends on the shape characteristics of crowns and the spectral response characteristics of leaves. However, without any interpretation of the response characteristics of leaves in the crowns, it is difficult to accurately isolate the shapes of the crown with various image segmentation methods due to the limited resolution of satellite images [18].

To express the relationships between spectral reflectance and forest structure parameters, statistical-based and physical-based models are widely applied to construct quantitative models [21–27]. Among them, statistical-based models are mainly focused on the correlations between remote sensing variables (spectral reflectance, vegetation indexes, and texture features) and forest parameters. These models include empirical statistical models and nonparametric models [8,28–33]. Forest parameters are normally estimated by these models due to the convenience and simplicity of model application. However, without considering any physical and chemical parameters of the forest, the strong geographical and spatiotemporal limitations make it difficult to achieve cross-

regional and inter-temporal predictions. In contrast, physical-based models, such as the Li–Strahler geometric–optical model and the SAIL, FLIGHT, and LESS radiative transfer model, are based on the physical principles of light transmission and can mathematically express the mechanisms between solar radiation and forest crowns [34–36]. Previous results demonstrated that forest structure information can be extracted from remote sensing images by simulating the optical properties of vegetation and the light transmission process by physical-based models [23,37–40]. Furthermore, previous studies also illustrated that the heterogeneity of intra-class pixels has high sensitivity with forest crown information, as demonstrated by simulating the optical properties of vegetation with physical models [41–44]. However, many input parameters (some of which may not be available) are required to describe the relationships between remote sensing variables and forest parameters. The lack of these parameters undoubtedly reduces the usefulness and generalizability of physics-based models. Therefore, to secure the advantages of statistical-based and physical-based models, a potential solution is to construct simple features with clear physical meaning based on physical-based models and apply them to map forest parameters using statistical-based models.

Additionally, the accuracy of mapping tree CD values is also determined by the methods used to extract features from remote sensing images [21,45]. It is very difficult to describe the shapes of the forest canopy with only a few neighboring pixels, particularly for high-spatial-resolution images [46]. Currently, filter-based and object-based methods are commonly used for feature extraction. Filter-based methods are often applied to reduce uncertainty of features related to the forest, and object-based methods are more often applied to obtain more homogeneous pixels [21,47]. However, it is difficult to choose appropriate feature extraction methods to obtain spatial and optical features from high-spatial-resolution images. Furthermore, the GF-2 satellite, the first Chinese civil satellite with sub-meter spatial resolution, has been successfully applied to map forest parameters in various regions [48]. However, few studies have focused on the performance of GF-2 images for extracting tree CDs.

The objective of this study is to interpret the relationships between the spatial heterogeneity of GF-2 images and tree CD values based on the 3D radiative transfer model. The specific research content of this study includes the following. (1) Simulated images based on GF-2 images parameters were initially obtained to demonstrate the relationships between spatial features and tree CDs. (2) Spatial heterogeneity was employed to express tree CD values in simulated images and GF-2 images. (3) Tree CDs were mapped in a boreal forest, combined with other types of features by filter-based and object-based methods. (4) We analyzed targets in remote sensing feature matching and extraction.

## 2. Study Area and Data

### 2.1. Study Area

The study was carried out in the Wangyedian forest farm, located in Inner Mongolia Province, China (as shown in Figure 1). The study area falls within the mid-temperate continental monsoon climate zone, with an average annual temperature and precipitation of 4.2 °C and 400 mm, respectively. The topography of the forest farm is mainly mountainous, with an altitude ranging from 500 m and 1890 m. The forest area in the study region is nearly 23,118 ha, and the main tree species of artificial forest in the study area are Chinese pine and larch.

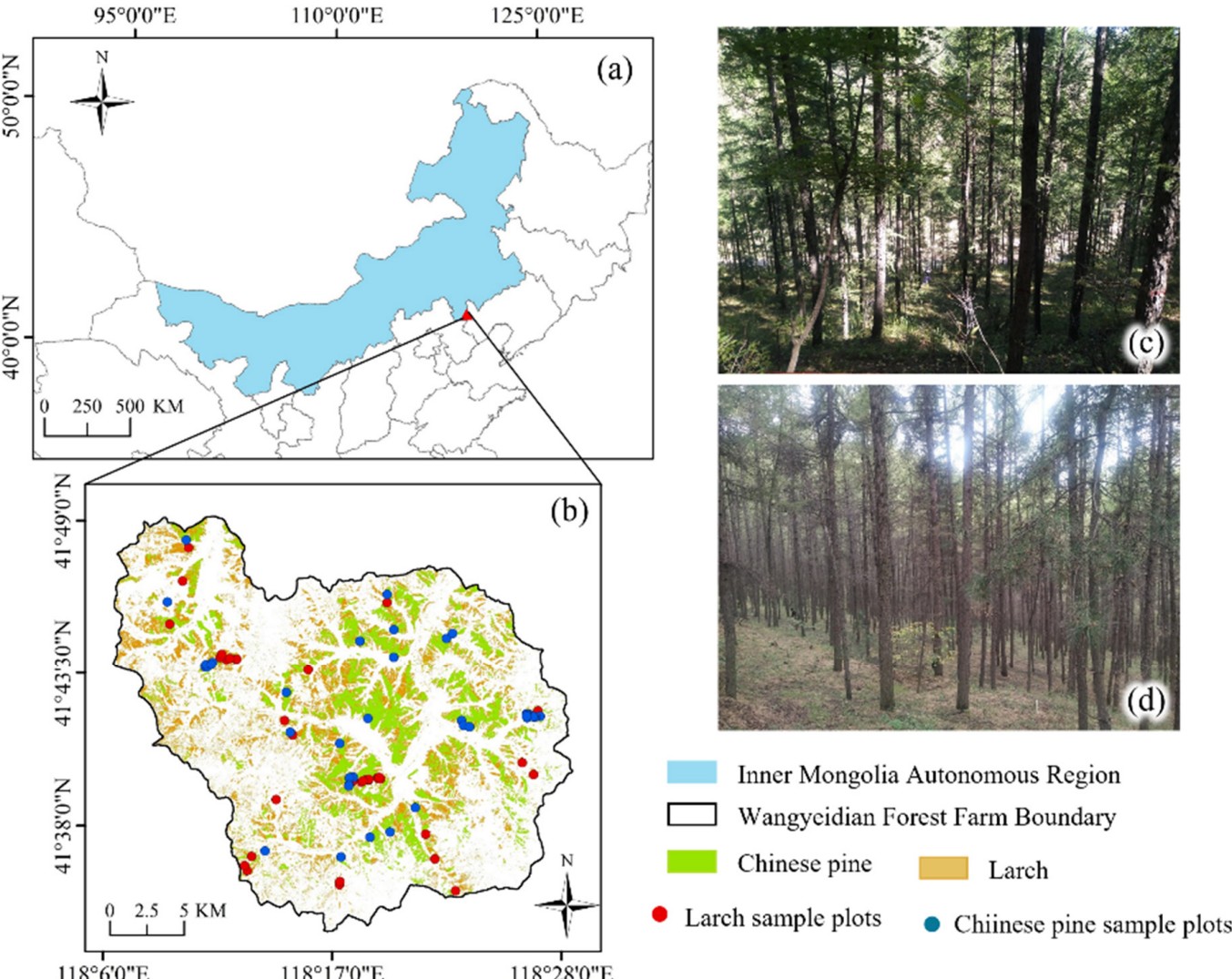

**Figure 1.** Location map of the study area: (**a**) is the boundary of Inner Mongolia Province, (**b**) is the boundary of the Wangyedian forest farm, (**c**) is a photo of Chinese pine in the sample plots, and (**d**) is a photo of larch in the sample plots.

### 2.2. Ground Data Collection

Based on the updated forest management inventories data in 2017, the tree species and area ratios within the forest were summarized, and the area and boundaries of coniferous species were obtained. Then, a total of 79 coniferous forest plots, including 37 larch plots and 42 Chinese pine plots, were randomly sampled, and field data collection was completed in October 2017. The global positioning system (GPS) was used to obtain the coordinates of the center and four corner points of each 25 × 25 m square plot. The diameter at breast height (DBH), height, and crown height (CH) of each tree were measured separately using a diameter ruler and a goniometer. The CD of each tree was measured separately in the east–west and north–south directions using a tape measure, and the average of two measured values was considered as the CD of each tree. Moreover, the density was obtained by dividing the number of trees in the plot by the area of the plot. The average value of the structural parameters of all trees in the plot is considered as in situ data. The results of ground-measured data are listed in Table 1.

**Table 1.** The statistical information of CD values in ground-measured samples.

| Tree Parameters | Range | Average Value | Coefficient of Variation (%) |
|---|---|---|---|
| DBH (cm) | 6.8–27.9 | 16.7 | 32.8 |
| Height (m) | 6.8–23.7 | 13.6 | 27.3 |
| CH (m) | 3.5–12.3 | 7.1 | 32.8 |
| CD (m) | 1.8–6.5 | 3.4 | 30.2 |
| Density ($10^3$/ha) | 0.26–6.5 | 1.7 | 70.1 |

### 2.3. Remote Sensing Data and Image Pre-Processing

The GF-2 satellite is China's first sub-meter spatial resolution civil remote sensing satellite, carrying a 1 m spatial resolution panchromatic camera and a 4 m spatial resolution multispectral (blue, green, red, and near-infrared) camera. In this study, two GF-2 images were applied to map the tree CD (http://www.cresda.com/CN/, accessed on 26 August 2017). Before extracting features from GF-2 images, pre-processing is essential to improve image quality. The pre-processing of GF-2 images was implemented in ENVI 5.3, including radiometric correction, atmospheric correction, image fusion, geometry correction, and topographic correction. Finally, the multispectral images with a 1 m spatial resolution were obtained to be employed to extract remote sensing features.

### 2.4. Interpretation of Spatial Heterogeneity

#### 2.4.1. Spatial Heterogeneity of Forest Crown

Commonly, each pixel in the images represents a spatial region that corresponds to the spatial resolution, and the spectral reflectance of a pixel is directly determined by the targets in the pixel and their spectral reflectance. Specifically, a pixel in low- and medium-spatial-resolution images may cover several different targets with different spectral reflectance. Based the radiative transfer model, the spectral brightness value of each mixed pixel can be expressed by four basic components, including lighted vegetation (*C*), shaded vegetation (*T*), lighted ground (*G*), and shaded ground (*Z*) [12]. Ultimately, the reflectance of each mixed pixel can be obtained from the spectral brightness values of four components by a linearly weighted combination, and the model can be expressed as:

$$R = \frac{1}{A}(K_C \times S_C + K_T \times S_T + K_G \times S_G + K_Z \times S_Z) \tag{1}$$

where $R$ is the spectral reflectance of the mixed pixel; $K_C$, $K_T$, $K_G$, and $K_Z$ are the spectral reflectance values per unit area caused by lighted canopy, shaded canopy, lighted background, and shaded background, respectively. $S_C$, $S_T$, $S_G$, and $S_Z$ are the areas of four components within pixel, and A is the area of a mixed pixel.

Normally, the surface of forest crown is considered as a complex medium in which the inclination of leaves at different positions is spatially heterogeneous. When the size of the pixels is much smaller than the size of the canopy, the reflectance of pixels caused by the illuminated canopy is naturally determined by the distribution of branches and leaves in the illuminated canopy. Based on the Phong lighting model [49], the reflection intensity of a leaf under illumination is directly dependent on the incident direction of the light and the inclination angle of leaves, and the model can be expressed as

$$Ld = kd\left(\frac{I}{r^2}\right)\max\left(0, \vec{n} \cdot \vec{l}\right) \tag{2}$$

where *Ld* is diffusely reflected light; *kd* is the coefficient of diffuse reflection; *I* is the intensity of the incident light; *r* is the propagation distance of light; $\vec{n}$ and $\vec{l}$ are the incident direction of light and the normal vector direction, respectively; and max indicates the rejection of light with an angle greater than 90. In addition, leaves and branches can be projected onto each other so that observers can observe the leaves in shadow, even on the lighted side.

Thus, based on the Formula (1), the spectral reflectance of a pixel within a lighted canopy can be expressed as

$$R = \frac{1}{A}(R_L \times S_L + R_S \times S_S + R_B \times S_B) \tag{3}$$

$$R_L = \frac{1}{S_L} \int_{\beta}^{\gamma} Ld(\alpha)d\alpha \tag{4}$$

where $R_L$ is the average spectral reflectance of the lighted leaves; $\gamma$ and $\beta$ are the maximum and minimum angle between the incident direction of light and the normal vector of leaves within a pixel, respectively. $S_L$ is the area of lighted leaves; $R_S$ and $S_S$ are the average spectral reflectance and area of shaded leaves, respectively. $R_B$ and $S_B$ are the spectral reflectance and the area of background, respectively. It is inferred that the spectral reflectance of the forest canopy is determined by both the canopy structure and the distribution of the leaves' inclination angle. Thereby, the diversity of inclination angles of leaves resulted in diversity in spectral reflectance within the canopy.

In statistics, the coefficient of variation is often used to evaluate the dispersion degree of data by eliminating the effects of the measurement scale and magnitude. For the images with a high spatial resolution, the spectral reflectance diversity of all pixels within a forest canopy can be expressed by spatial heterogeneity using the coefficient of variation. Within the instantaneous field of view, the spatial heterogeneity (SH) of pixels can be obtained as follows:

$$SH = \sqrt{\frac{\sum_{i=1}^{n}\left(R_i - \overline{R}\right)^2}{n}} \Big/ \overline{R} \tag{5}$$

where $R_i$ is the spectral value of the pixel, $\overline{R}$ is the average of the spectral values of the pixels within a scene, and $n$ is the total number of pixels within the scene.

### 2.4.2. The Response of Spatial Heterogeneity with CD Values

In a uniformly growing coniferous plantation forest, each tree can be viewed as a geometric structure consisting of cones and cylinders (Figure 2a). By imaging the simplified coniferous plantation forest scene with a high spatial resolution (1 m), the projection of the coniferous forest crown can be obtained, as shown in Figure 2b,c, and the projection of the crown can be reduced to combination of a circle and a triangle that gradually brightens towards the center and a shaded triangle.

Based on Formula (4), an extreme difference in the leaves' inclination within the crown is expressed as follows:

$$\sigma = \gamma - \beta \tag{6}$$

The gradient of the pixel values in one direction is defined as:

$$n = \frac{\theta_{max} - \theta_{min}}{\sigma} = \frac{CD}{2 \times M} \tag{7}$$

where $\theta_{max}$ and $\theta_{min}$ are the maximum and minimum values of the leaves' inclination angle in the crown, and the extreme difference in all pixels within crown can be modeled as:

$$Max.R_L - Min.R_L = \frac{1}{S_L} \int_{\theta_{max}-\sigma}^{\theta_{max}} Ld(\alpha)d\alpha - \frac{1}{S_L} \int_{\theta_{min}}^{\gamma\theta_{min}+\sigma} Ld(\alpha)d\alpha \tag{8}$$

Normally, the fineness of the forest crown depends on the number of pixels within the crown and the spatial resolution of images. For images with a certain spatial resolution, a large crown has more pixels, and an increase in the extreme difference can enrich the diversity of spectral reflectance within the crown. It is inferred that the spatial heterogeneity is proportional to the size of the forest crown in images with a certain spatial resolution.

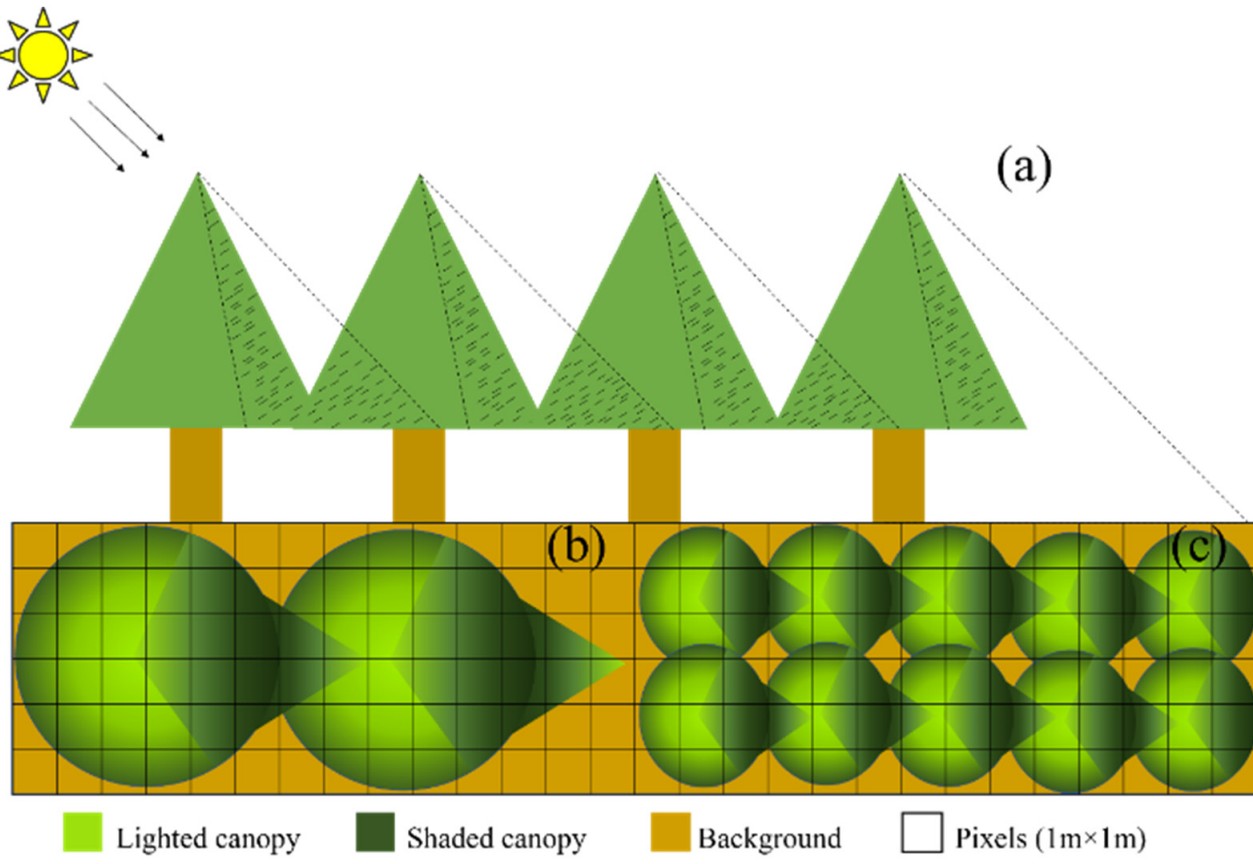

**Figure 2.** Schematic diagram of the geometric-optical model of the coniferous forest crown: (**a**) is a side view of the simulated coniferous forest, and (**b**,**c**) are projections of the crown with different sizes.

### 2.4.3. Simulated Images of Spatial Heterogeneity in Forest Ecosystems

The widely used large-scale remote sensing data and image simulation framework (LESS) model is a raytracing-based 3D real structural radiative transfer model that can simulate reflection characteristics from the leaf scale to the crown scale and output the corresponding simulation data such as spectral reflectance [34]. To further verify the relationships between spatial heterogeneity and forest crown, simulated images with the same wavelength as the GF-2 images were generated based on the LESS model. Moreover, in this study, the size of the simulated image was $100 \times 100$ m and the spatial resolution was 1 m.

### 2.5. *Variable Extraction and Selection*

### 2.5.1. Variable Extraction Methods

For the images with a high spatial resolution, the size of pixel is much smaller than the tree CD, and the target should be completely expressed by several pixels. To demonstrate the information of forest crown, these pixels involved in one crown should be exactly derived from the images [10]. Recently, two methods, filter-based and object-based, are employed to obtain these pixels [10,50]. Figure 3 illustrates the variable extraction methods (filter-based and object-based). It is inferred that the completeness of the forest crown depends on the size of the filter-based method and the segmentation scale of the object-based method, respectively. In this study, various sizes and various segmentation scales were used to obtain alternative variables. Then, the sensitivity between the extracted variables and the tree CD was evaluated to obtain the optimal size and segmentation scale of two methods.

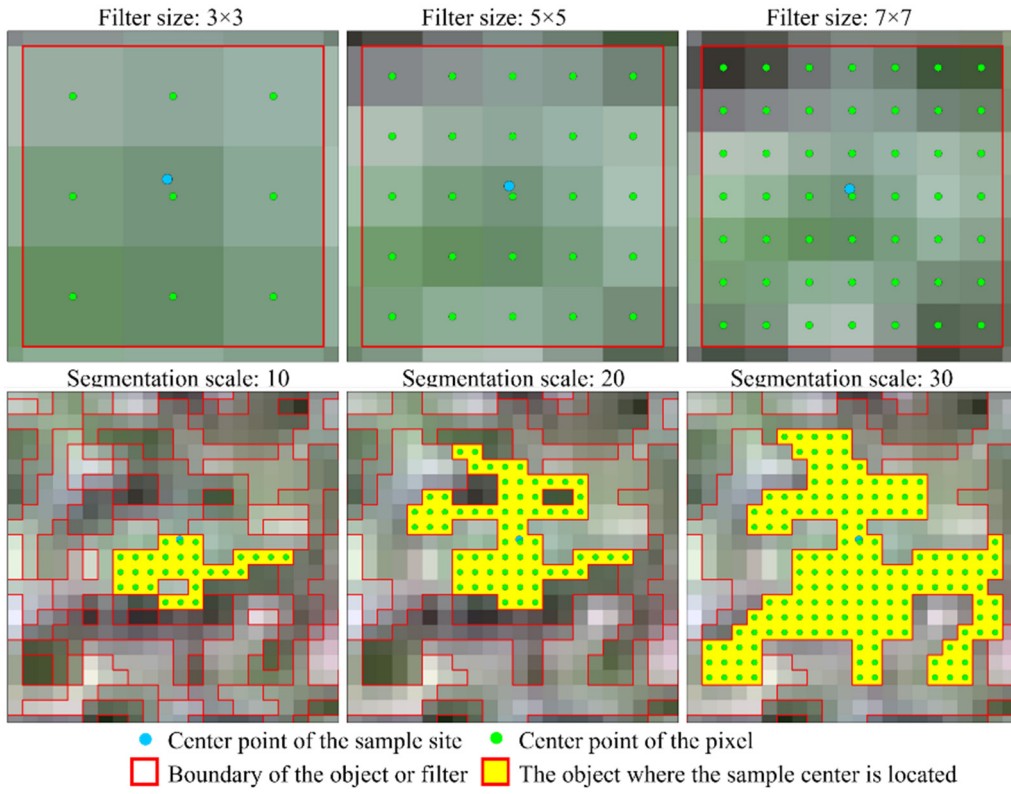

**Figure 3.** Variable extraction using filter-based and object-based methods.

### 2.5.2. Variables Extracted from GF-2

In this study, three variable sets (spectral features, texture features, and spatial heterogeneity) were extracted from GF-2 images to map CD values. Among them, widely used spectral features, including the surface reflectance of four multispectral bands and nine vegetation indices, were initially extracted from pre-processed images, including the normalized difference vegetation index (NDVI), the soil-adjusted vegetation index (SAVI), the red–green vegetation index (RGVI), the enhanced vegetation index (EVI), the modified soil-adjusted vegetation index (MSAVI), the difference vegetation index (DVI), the modified simple ratio (MSR), the ratio vegetation index (RVI), and the atmospherically resistant vegetation index (ARVI) [7,30–32]. Furthermore, texture features extracted by gray-scale coevolution matrices (GLCMs) are considered important visual cues of remote sensing images [9,32,51]. In this study, for each band and vegetation index, eight texture features with different sizes and segmentation scales were applied to express the forest crown, including the mean (ME), variance (VAR), homogeneity (HOM), contrast (CON), dissimilarity (DIS), entropy (ENT), second moment (SM), and correlation (COR). In addition, based on the geometric optical model, it is inferred that spatial heterogeneity has great potential to express forest crown areas. So, the spatial heterogeneity with various sizes and segmentation scales was extracted from each band and vegetation index.

### 2.5.3. Variable Selection and Combination

To map tree CD values using these alternative variables, the accuracy of the results significantly depends on the variable selection methods and the optimal variable set. In complex forest environments, a suitable variable selection method can reduce the influence of many alternative variables and improve the performance and efficiency of the model. Based on the random forest (RF) algorithm, the widely used Boruta algorithm creates a shadow feature matrix based on the original feature matrix, and then compares the importance of the original features with the shadow features [52]. The optimal feature set is ultimately obtained by eliminating any irrelevant features. In this study, optimal feature sets of each type, including spectral features (SFs), texture features (TFs), and

spatial heterogeneity (SH), were obtained using the Boruta algorithm with two different features extraction methods. Then, combinations with different strategies were performed to compare the feature extraction methods. The detailed optimal feature sets are listed in Table 2.

**Table 2.** The lists of variable sets.

| Number | Feature Type | Filter-Based | Objected-Based |
|---|---|---|---|
| 1 | Spectral features | SFs | SFs |
| 2 | Texture features | TFs | TFs |
| 3 | Spatial heterogeneity | SH | SH |
| 4 | Combinations within feature extractions | SFs + SH | SFs + SH |
| 5 | Combinations within feature extractions | SFs + TFs | SFs + TFs |
| 6 | Combinations between feature extractions | SFs (objected) and SH (filter) | |
| 7 | Combinations between feature extractions | SFs (objected) and TFs (filter) | |

*2.6. Models of Mapping Tree CD Values and Assessment*

Recently, machine learning algorithms have been widely used for estimating forest parameters [53]. To map forest crowns, three machine learning algorithms (SVMs, RF, and KNN) and one traditional regression model (MLR) were employed to predict CD values. In this study, the construction and optimization of each model were implemented in the R software. Moreover, the coefficient of determination ($R^2$), root mean square error (RMSE,) and relative root mean square error (rRMSE) derived from leave-one-out cross-validation were employed to evaluate the performance of the models.

## 3. Result

*3.1. Spatial Heterogeneity Extracted from Simulated Images*

To simulate optical images with the LESS model in forest ecosystems, the structural parameters of trees, including the CD, DBH, height, crown height, and density, are important in order to construct the simulation scenarios. The stereoscopic model of trees with different CD values was established according to the anisotropic growth equation, and the density was used to determine the number of trees within each scene. Trees were randomly placed inside the scene, and the distance between trees was set to be greater than 0.5 times of the CD. Based on the wavelengths of GF-2 bands, simulated images of different tree CD values are shown in Figure 4.

Figure 4 illustrates that the ability of recognizing tree CD values in images depends on the size of the CD. In simulated images with small tree CD values (1 m and 2 m), almost all pixels were mixed pixels, consisting of light and shadow scenes, making it difficult to recognize the lighted crown and lighted background. As the size of the tree CD increased, the shapes of crown became visible and the pixel difference between the lighted crown and lighted background obviously increased.

To test the relationships between spatial heterogeneity and tree CD, the spatial heterogeneity of various CD sizes (ranging from 1 m to 7 m) were extracted from each simulated band using a filter-based feature extraction method with various sizes (ranging from 3 to 31). The relationships between the spatial heterogeneity, bands, and sizes of CD are illustrated in Figure 5. For four simulated bands, the values of spatial heterogeneity were significantly correlated with the size of the tree CD, and the spatial heterogeneity values increased with the increase in the tree CD in each band. The results also confirmed that the spatial heterogeneity could effectively map tree CD values because of the proportionality to the sizes of the tree CD.

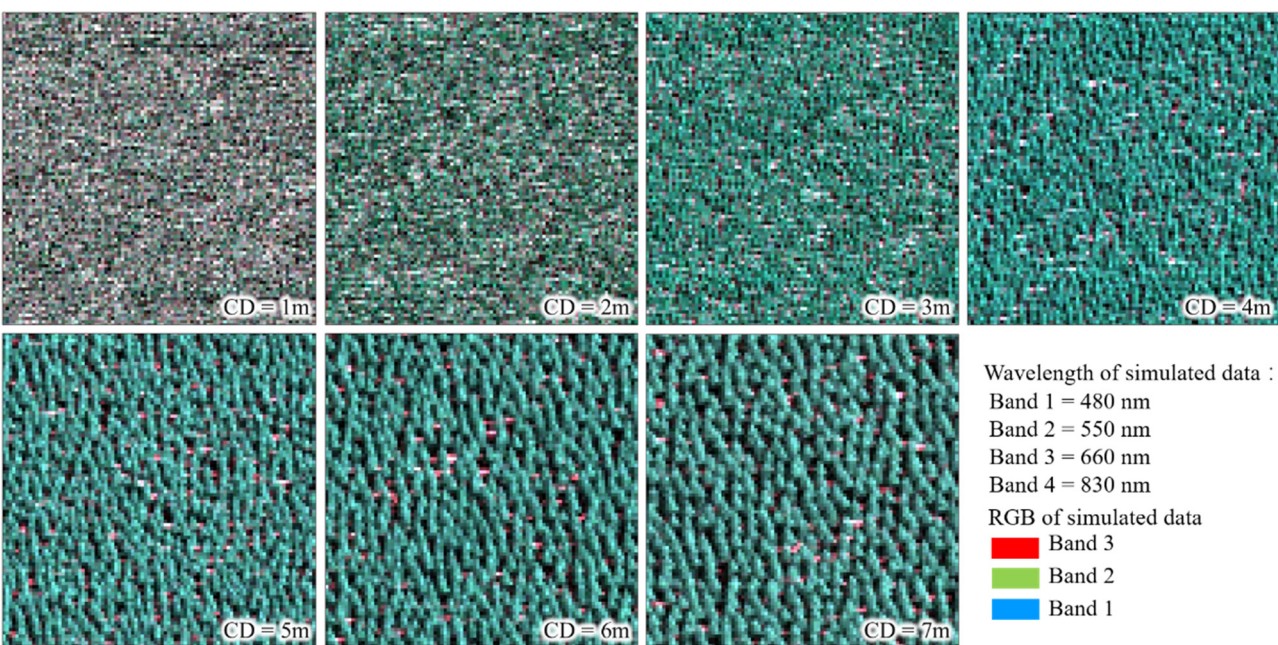

**Figure 4.** Simulated multispectral images with various sizes of CDs based on the LESS model.

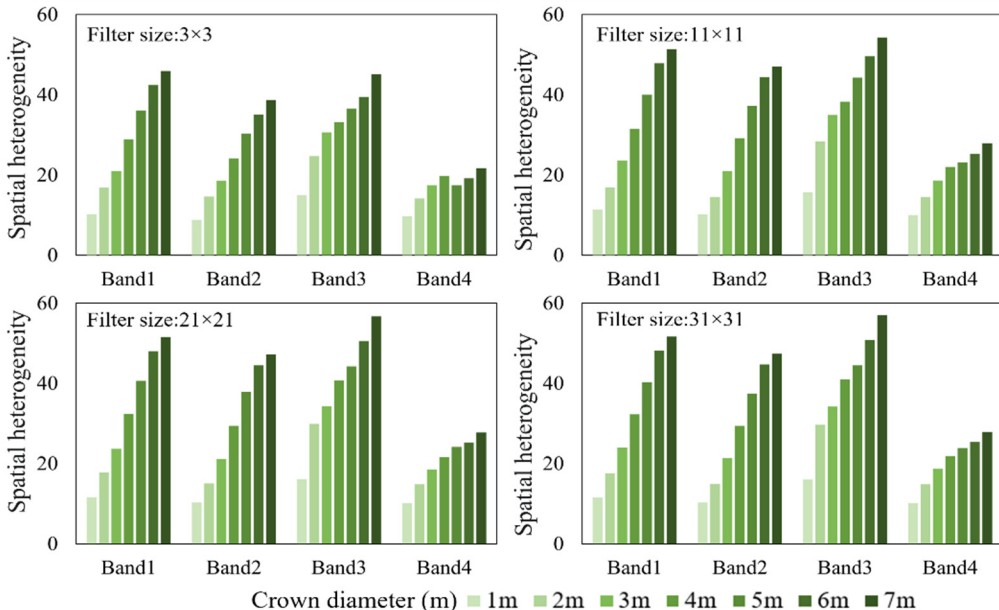

**Figure 5.** Relationships between the CD and SH of simulated images.

Furthermore, the spatial heterogeneity also varied with the different sizes of the filter-based method, and the plots of spatial heterogeneity with various windows sizes are illustrated in Figure 6. It can be found that spatial heterogeneity varied with the window sizes when the filter size was smaller than the tree CD. After the window size was larger than the tree CD, the spatial heterogeneity remained constant. The results of simulated images demonstrated that the reliable spatial heterogeneity should be obtained by appropriate window sizes in relation to the CD.

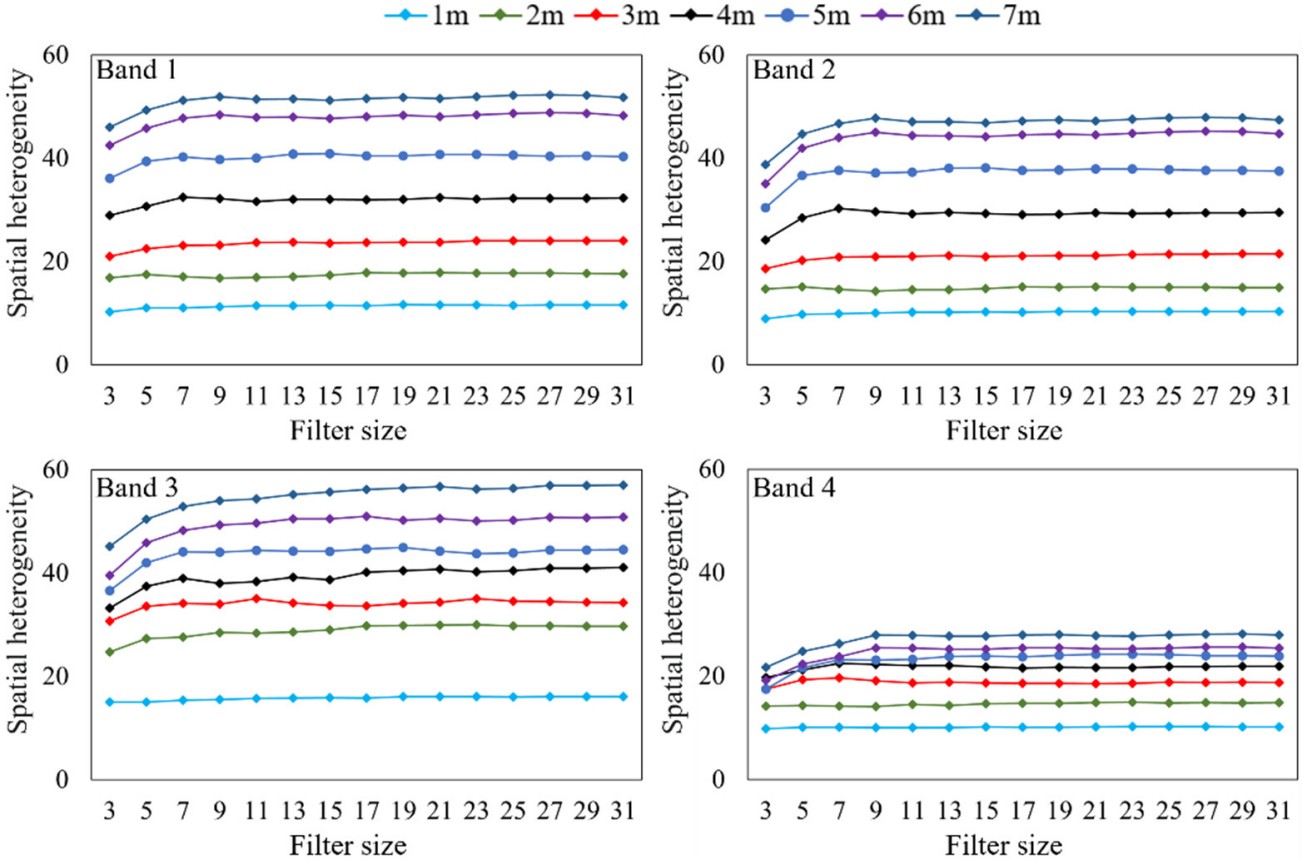

**Figure 6.** The SH plots varied with the window sizes of the filter-based method.

*3.2. The Results of Spatial Heterogeneity Related with the CD*

To further verify the relationships between the SH and tree CD, GF-2 images with four bands were employed to obtain three types of variables using the filter-based method with window sizes ranging from 3 to 31 and objected-based methods with segmentation scales ranging from 10 to 100, respectively. Based on the sizes of the tree CD, statistical graphs of SH extracted from four bands are illustrated in Figure 7.

Similar to the simulated images, correlations between the tree CD and SH were also found for GF-2 images. The results show that the SH was highly dependent on bands, the variable extraction methods, windows sizes, or segmentation scales and sizes of the tree CD. Because of the difference in response between bands and forest structure parameters, the SH values of the red band, green band, and blue band were larger than that from near-infrared bands for all tree CD sizes. The results also confirmed that the CD sizes were positively related to the SH values. A larger CD implies a more complex distribution of branches and leaf inclination angles, increasing the variance of pixels within the crown. It is inferred that SH has great potential to map tree CD values.

In addition, the gaps of SH values were also induced by variable extraction methods. Figure 8 also illustrates that the derived SH values were significantly dependent on the sizes of windows and segmentation scales for two different methods. For the filter-based method with various window sizes, when the size of the window was smaller than the CD, the SH value changed with the sizes of the extracted window. Once the size of the window was larger than the CD, the SH value changed very weakly. For the objected-based method, the SH values varied little with segmentation scales, because of the lower heterogeneity of the pixels inside the object.

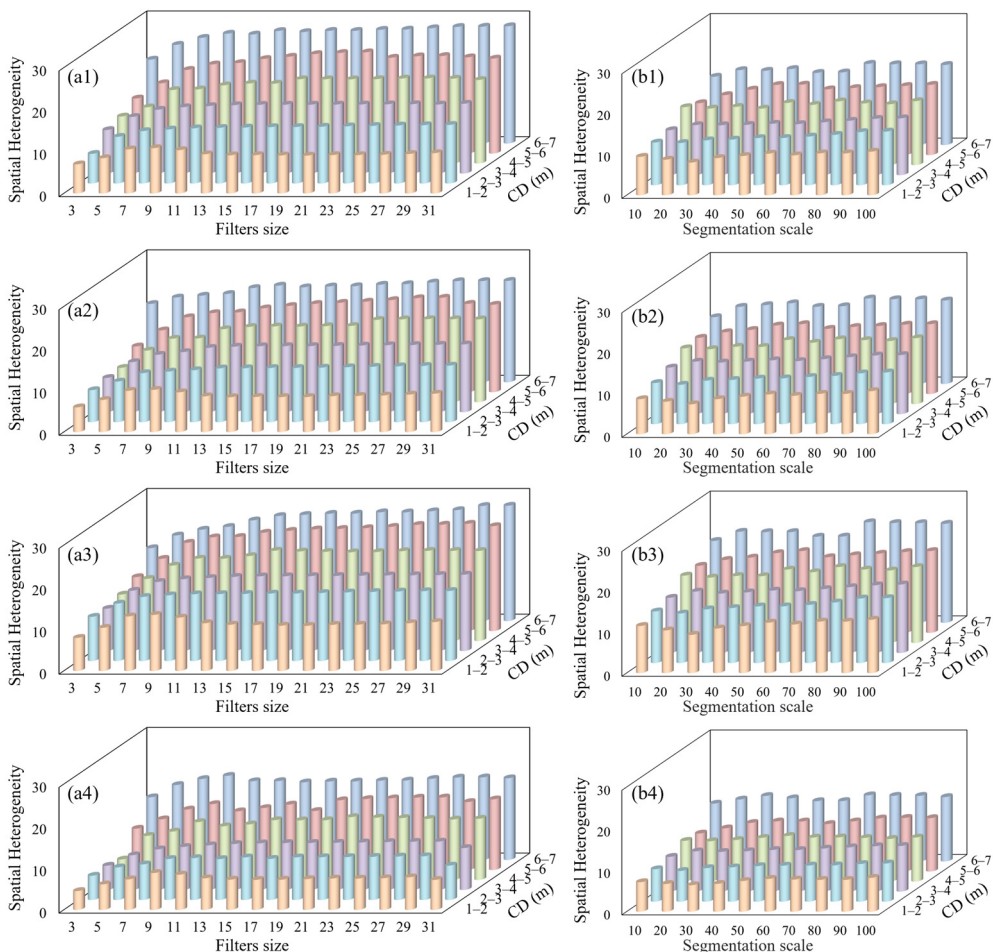

**Figure 7.** Statistical graphs of SH with different tree CD sizes extracted from GF-2 using two variable extraction methods: (**a1**–**a4**) statistical graphs of SH extracted from band1, band2, band3, and band4 using filter-based method with window sizes ranging from 3 to 31, respectively; (**b1**–**b4**) statistical graphs of SH extracted from blue, green, red, and NIR bands using objected-based methods with segmentation scales ranging from 10 to 100, respectively.

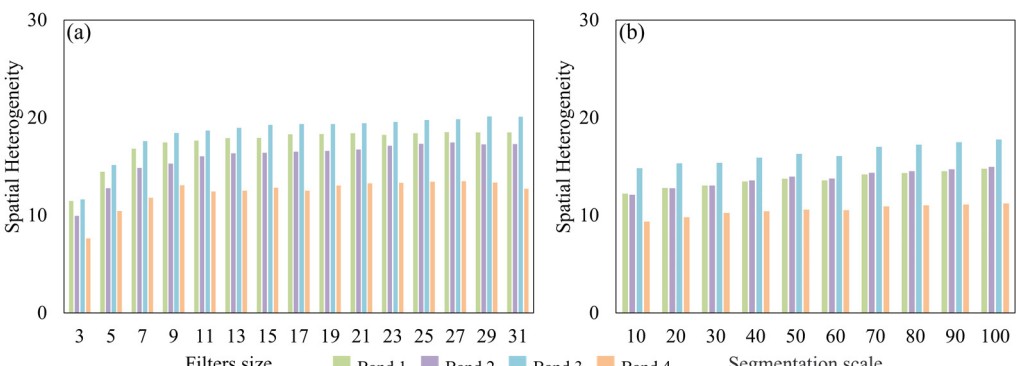

**Figure 8.** The results of SH using various window sizes (**a**) and segmentation scales (**b**).

### 3.3. Sensitivity between Tree CDs and Variables

To further evaluate the sensitivity between the CD and spatial heterogeneity, several types of variables, such as spectral bands, vegetation indices, and textural features, were extracted from GF-2 images using filter-based methods with various window sizes and object-based methods with various segmentation scales. Then, these features of spatial heterogeneity were also obtained from spectral bands and vegetation indices using two

variable extraction methods. The Pearson correlations between extracted variables and tree CDs are shown in Figure 9.

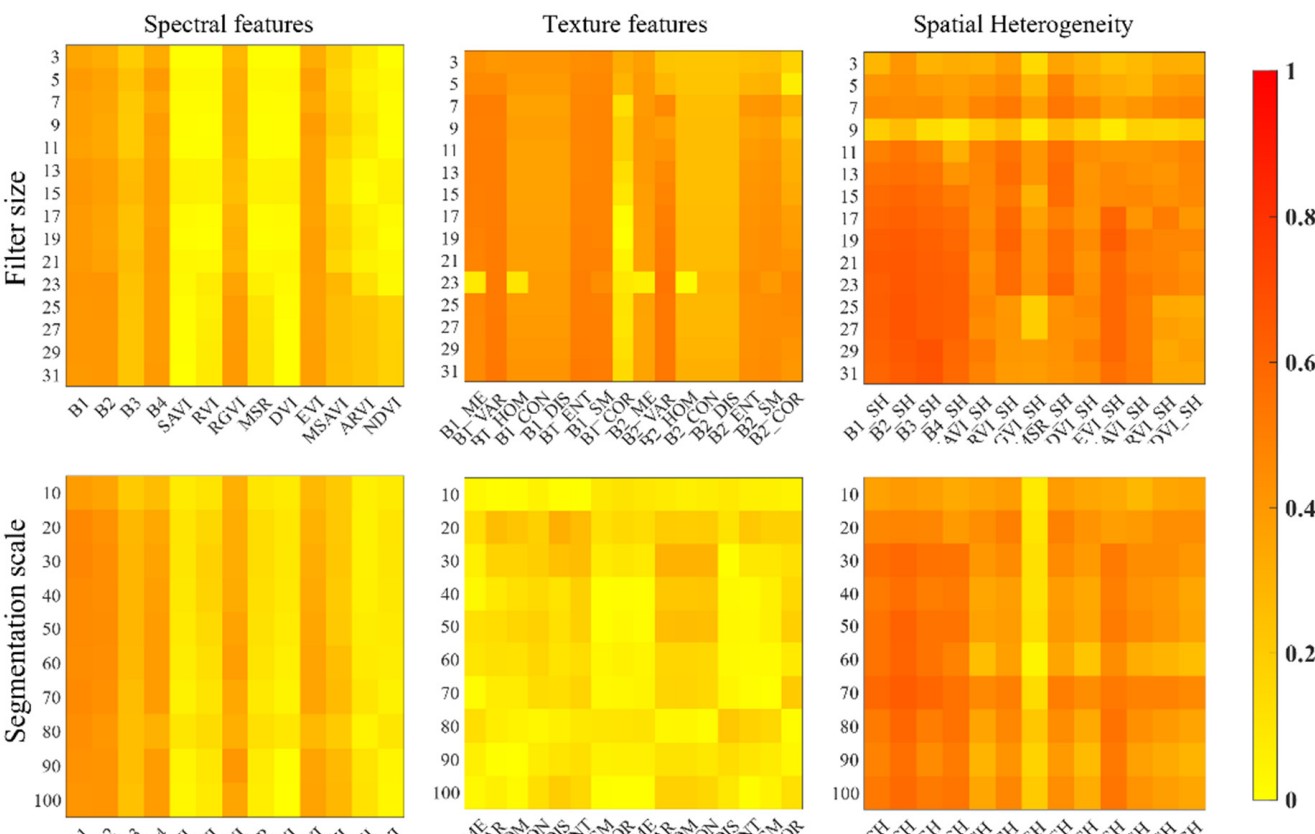

**Figure 9.** Pearson correlations between tree CDs and various types of variables.

Figure 9 illustrates that features of spatial heterogeneity were more sensitive to tree CD values than traditional spectral features and textural features. The Pearson correlations between spectral features and tree CDs ranged from 0.05 to 0.44 for the filter-based feature extraction method, and the correlation values between textural features and CD ranged from 0.17 to 0.5 for the filter-based feature extraction method. Moreover, the correlation values between textural features and CD extracted by object-based were less than 0.2. The Pearson correlations between SH and tree CDs ranged from 0.2 to 0.71 for the filter-based feature extraction method and from 0.1 to 0.62 for the object-based feature extraction method. The highest correlations of SH extracted from the green band were 0.71 using the filter-based method and 0.62 using the object-based method. The results demonstrated that features of spatial heterogeneity with high sensitivity could more effectively map tree CDs.

In addition, for the features of spatial heterogeneity, the values of Pearson correlation were also related to the sizes of filter windows. When the size of the filter window was less than 11, the correlation between spatial heterogeneity and tree CD varied with the sizes. After that, the values of Pearson correlation gradually increased with the sizes of filter windows. In contrast, the segmentation scale hardly affected the correlation between the variables and the CD. Furthermore, for this, the sensitivities between the spectral features and the tree CD extracted by the object-based method were slightly higher than those extracted by the filter-based method. Among the spectral features, the red band, green band, near-infrared band, RGVI, and EVI extracted by two method showed high correlations with CDs (ranging from 0.3 to 0.4).

To further analyze the sensitivity between the SFs, TFs, SH, and CDs, the scatterplots between the CDs of all samples and SFs, TFs, and SH extracted by the green band are illustrated in Figure 10. It is clear that the features of spatial heterogeneity showed a linear positive correlation with the CD for both filter-based and object-based methods, and their correlation coefficients were 0.71 and 0.62, respectively. Comparing with SFs and TFs, the SH can effectively construct the models for mapping tree CD values.

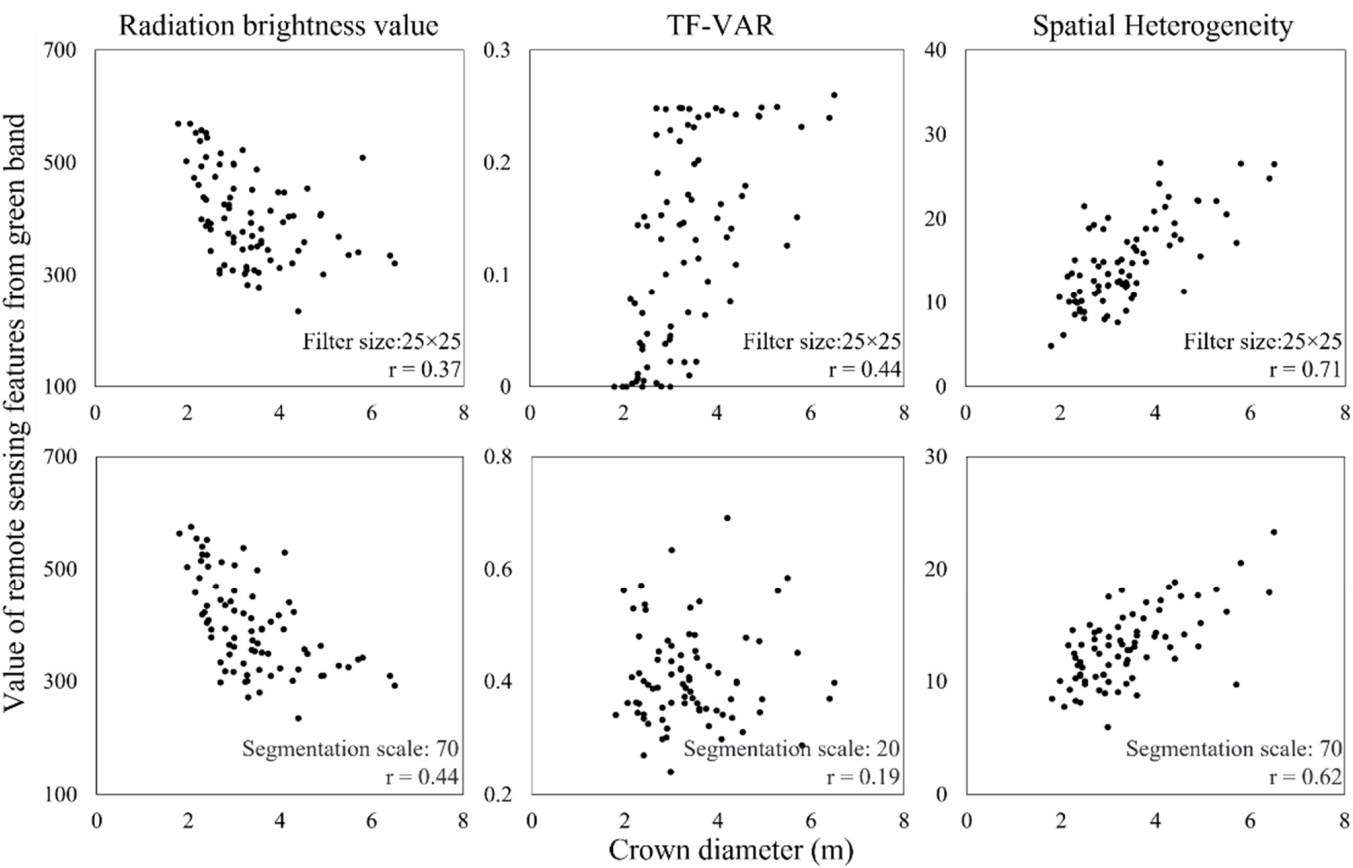

**Figure 10.** Scatter plots of CDs and partial variables extracted from the green band.

*3.4. Results of Mapped CDs Using GF-2 Images*

To further verify the spatial heterogeneity performance in mapping CD values, four regression algorithms (RF, KNN, SVR, and MLR) were applied to establish the models between each optimal variable set and the CD. In this study, all extracted features were grouped into three categories: spectral features (SFs), texture features (TFs), and spatial heterogeneity (SH). Then, several combinations with different strategies were also formed within one feature extraction method (SFs + TFs and SFs + SH) and between different feature extraction methods. The Boruta algorithm was employed to obtain the optimal variable set for each type of alternative feature set. Figure 11 illustrates the accuracy indices of mapped tree CD using various optimal variable sets by filter-based and object-based methods.

The results found that accuracy indices (R2 and rRMSE) varied with sizes of filter windows and segmentation scales, especially for spatial heterogeneity and combined variable sets. In three single types of features, variations in mapping tree CD values with filter window sizes and segmentation scales were smaller for spectral features than those for texture features and spatial heterogeneity, and the accuracy of results when using the object-based method was slightly higher than that when using the filter-based method. Moreover, the ability of spatial heterogeneity to map tree CD values significantly improved after obtaining stable features with appropriate filter window sizes. Even for the size larger

than $17 \times 17$, the spatial heterogeneity ability was obviously stronger than spectral features in mapping tree CD values. However, the results also illustrate that the ability of spatial heterogeneity to map tree CD values was weaker when using the object-based method than that when using the filter-based method. It is inferred that the filter-based method with appropriate sizes of filter windows was suitable for extracting spatial heterogeneity in mapping tree CD values.

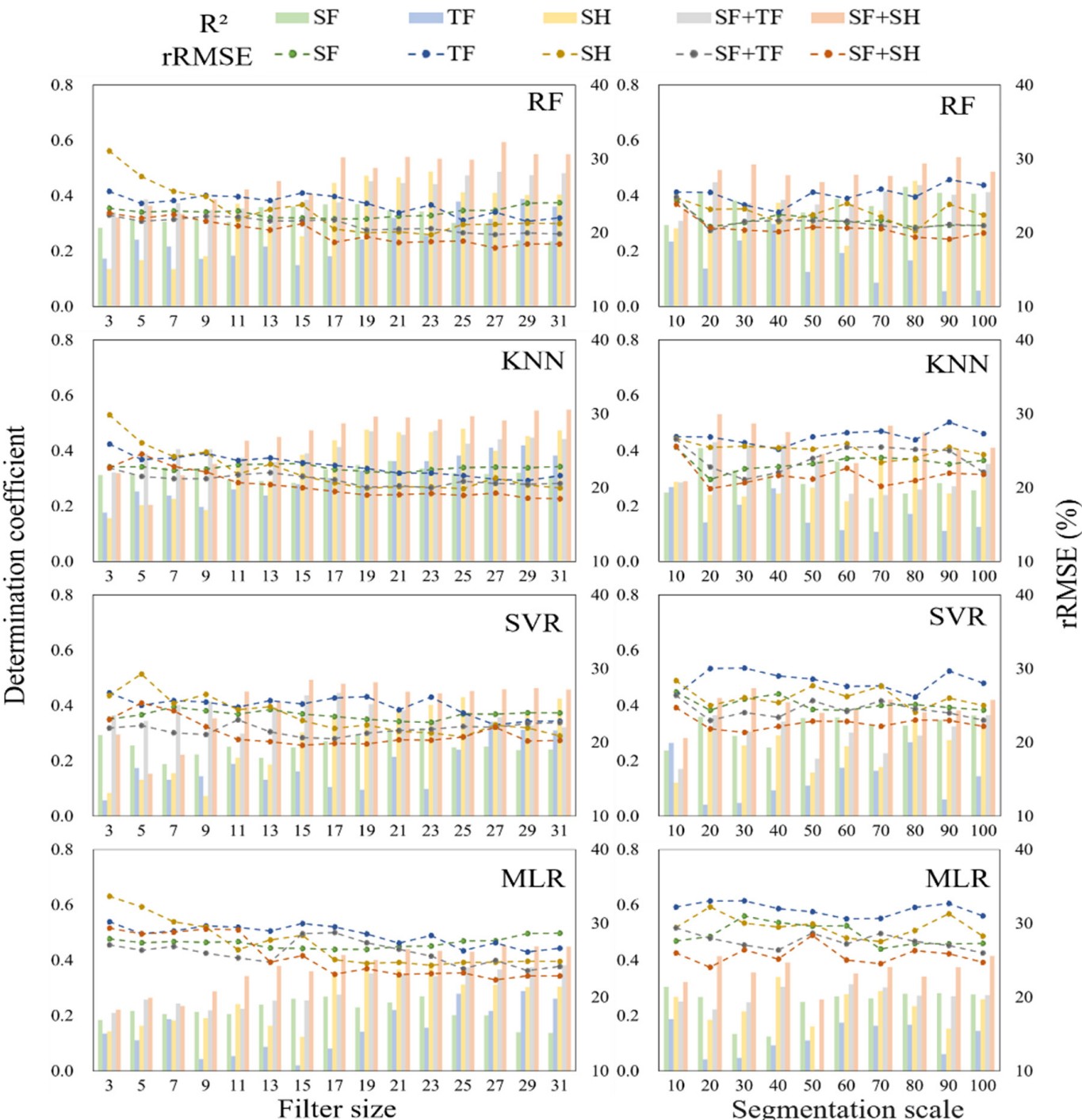

**Figure 11.** The accuracy of mapped tree CD values using four models with different feature extraction methods and their involved parameters.

Furthermore, the combined variable sets (SFs+TFs and SFs+SH) also showed great potential to increase the accuracy of mapping tree CD values. For each size of filter window and each segmentation scale, the values of $R^2$ obtained by combined variable sets were significantly larger than those obtained by single types of features (SFs, TH, and SH). After obtaining stable features of spatial heterogeneity, the optimal accuracy indices ($R^2$ and rRMSE) were mainly obtained from the combined variable set (SFs+SH). The results demonstrated that spatial heterogeneity was more sensitive to changes in the CD and was more effective in mapping CD values than texture features. Furthermore, the combination of spectral features and spatial heterogeneity (SFs+SH) had the most potential to express the relationships between remote sensing images and forest crowns.

To further analyze the contribution of SFs, TFs, and SH in mapping tree CD values with two feature extraction methods, the optimal results for each variable set are summarized in Table 3. In three single types of features, the $R^2$ values ranged from 0.37 to 0.49 using the filter-based method and from 0.29 to 0.44 using the object-based method. Moreover, the optimal results were obtained from the feature set of spatial heterogeneity (SH), and rRMSE values were 19.70% for the filter-based method and 20.30% for the object-based method. Moreover, when using combined variable sets within one feature extraction method (SFs+TFs and SFs+SH), the rRMSE values ranged from 17.88% to 19.70% for the filter-based method and from 18.39% to 20.00% for the object-based method. Furthermore, the optimal results were obtained from the combination of spectral features and spatial heterogeneity (SFs+SH). The results also demonstrated that the accuracy of the mapped tree CD obviously improved using combined variable sets between different feature extraction methods, and the $R^2$ and rRMSE values of optimal results ranged from 0.52 to 0.56 and from 15.76% to 16.68%, respectively. It is inferred the accuracy of mapping tree CD can be improved using the combination of spectral features extracted by the object-based method and spatial heterogeneity extracted by the filter-based method.

**Table 3.** Optimal estimation results for each optimal variable set.

| Feature Extraction Method | Data Set | Model | Size or Scale | Accuracy Indices | | |
|---|---|---|---|---|---|---|
| | | | | $R^2$ | RMSE (m) | rRMSE (%) |
| Filter-based | SFs | RF | 25 | 0.37 | 0.72 | 21.82 |
| | TFs | KNN | 25 | 0.41 | 0.70 | 21.21 |
| | SH | KNN | 25 | 0.49 | 0.65 | 19.70 |
| | SFs and TFs | RF | 25 | 0.49 | 0.65 | 19.70 |
| | SFs and SH | RF | 25 | 0.59 | 0.59 | 17.88 |
| Object-based | SFs | RF | 70 | 0.44 | 0.68 | 20.61 |
| | TFs | RF | 20 | 0.29 | 0.75 | 22.73 |
| | SH | RF | 70 | 0.43 | 0.67 | 20.30 |
| | SFs and TFs | RF | 40 | 0.45 | 0.66 | 20.00 |
| | SFs and SH | RF | 70 | 0.55 | 0.61 | 18.39 |
| Combination | SFs (object) and TFs (filter) | RF | Object (80) and filter (25) | 0.60 | 0.56 | 16.68 |
| | SFs (object) and SH (filter) | RF | Object (70) and filter (25) | 0.66 | 0.52 | 15.76 |

Using different variable sets with filter-based and object-based methods, scatterplots and residuals between predicted and ground-measured tree CD values are illustrated to further verify the potential of spatial heterogeneity (Figure 12). By using both single types of features and combined variable sets within one feature extraction method (SFs + TFs and SFs + SH), underestimated results were mainly found for samples with large CDs. This phenomenon was significantly corrected by using combined variable sets between different feature extraction methods. It is confirmed that selecting appropriate feature extraction methods and involved parameters for different types of features is helpful to improve the accuracy of mapping tree CD values.

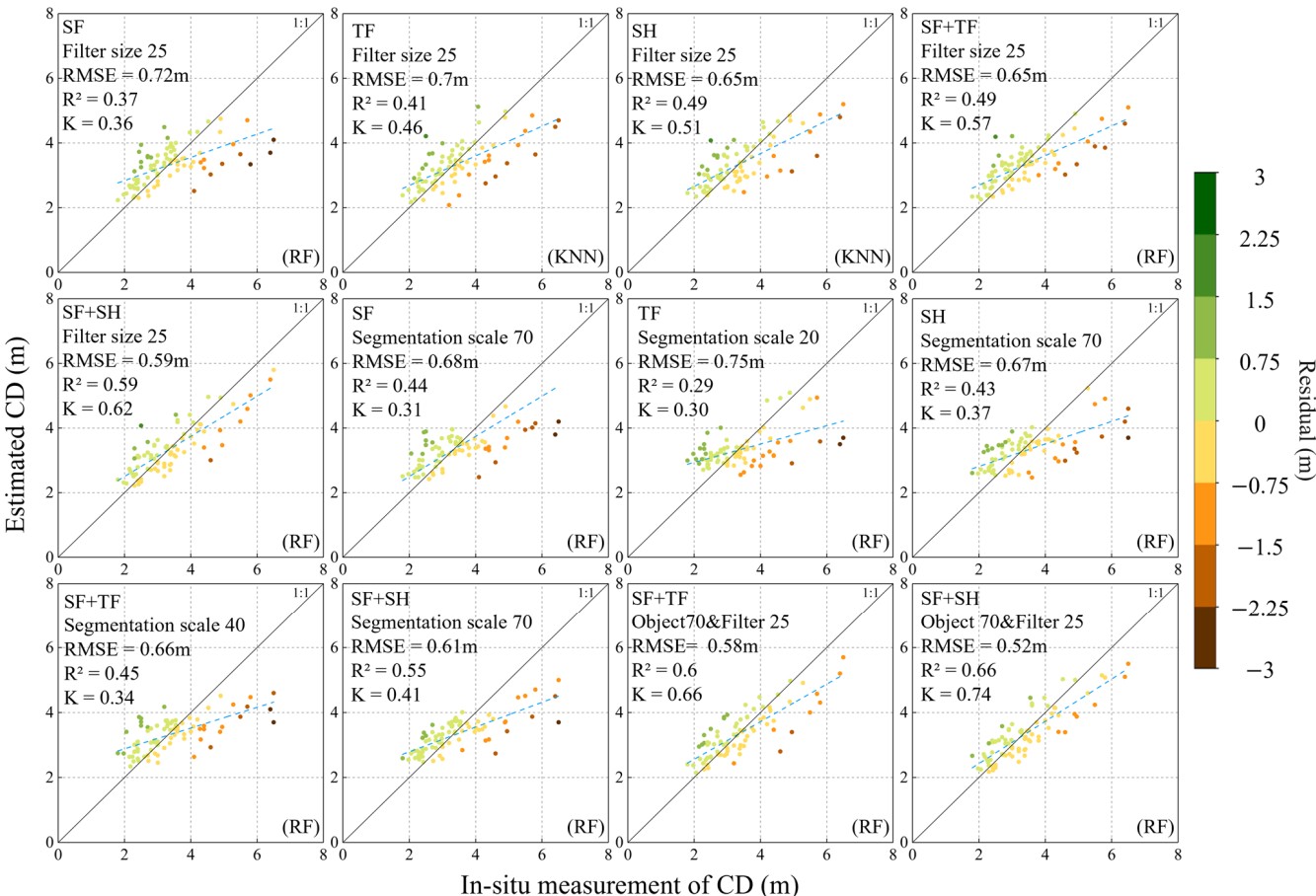

**Figure 12.** Scatterplots between predicted and ground-measured tree CDs using different variable sets with filter-based and object-based methods (the blue dashed line is the fitted line, the color of points is determined by the residuals between the predicted and ground-measured tree CD values, and K is the slope of the fitted line).

Based on the optimal results listed in Table 3 and Figure 12, the combination sets with different strategies, within one feature extraction method (SFs + TFs and SFs + SH) and between different feature extraction methods (SFs (object) and TFs (filter) and SFs (object) and SH (filter)), were applied to map tree CD values in planted boreal forest ecosystems (Figure 13). For the maps obtained from two combination variable sets with the filter-based method (Figure 13(a2,b2)), the "salt and pepper" phenomenon frequently occurred, which is obviously unreliable for a uniformly growing planted forest. In contrast, more rational maps of CD values were obtained from combination variable sets with the object-based method (Figure 13(c2,d2)). Considering the accuracy and rationality of maps, it is demonstrated that the combination sets between different feature extraction methods (SFs (object) and TFs (filter) and SFs (object) and SH (filter)) can help to reasonably and accurately obtain a map of tree CD values (Figure 13(e2,f2)).

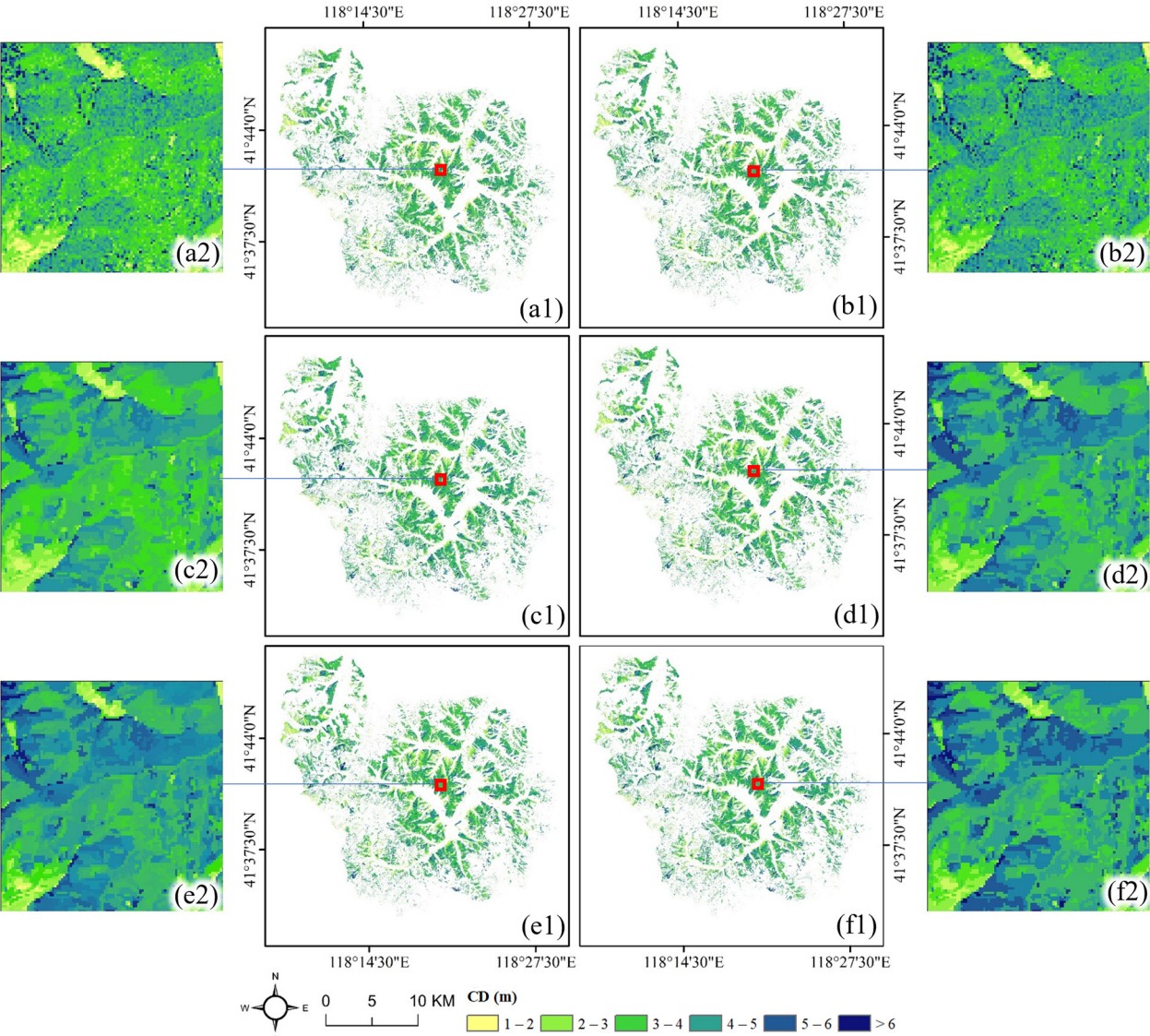

**Figure 13.** The maps of tree CD values using combined variable sets: (**a1,b1**) are combined variable sets (SFs + TFs and SFs + SH) with the filter-based method, respectively; (**c1,d1**) are combined variable sets (SFs + TFs and SFs + SH) with the object-based method, respectively; (**e1,f1**) are combined variable sets (SFs (object) and TFs (filter) and SFs (object) and SH (filter)) extracted with object-based and object-based metods, respectively; (**a2–f2**) are enlarged maps of corresponding local areas, respectively.

## 4. Discussion

### 4.1. The Relationships between Spatial Heterogeneity and Forest Crowns

Generally, the accuracy of detecting and depicting the shapes of forest crowns is highly related to the spatial resolutions of employed images [2]. As the spatial resolution of the available satellite images increases, more detailed information of forest crowns becomes visible. Previous studies mainly focused on high-spatial-resolution images acquired by aerial photographs and optical satellite images with sub-meter levels using various segmentation methods for images [16,47]. Limiting the cost of acquired images, satellite-based meter-level images are more cost-effective and suitable for mapping forest parameters in large regions. Compared with sub-meter images, it is more difficult to accurately isolate the shapes of the crown by image segmentation. Therefore, the spectral reflectance of pixels in the crown should be considered in expression forest crowns [18].

In our study, GF-2 images with a 1 m spatial resolution were acquired and the illumination differences within crowns could be recognized by pixels covered in the crown. Then, the spatial heterogeneity of spectral reflectance between certain number of pixels in crown was applied to express the diversity of the inclination angles of leaves. Compared with spectral features and texture features, it is inferred that the spatial heterogeneity of spectral reflectance has more potential to describe the changes in forest crown sizes based on simulated and GF-2 images. The results also demonstrate that spatial heterogeneity was highly correlated with CDs (r > 0.7) and increased with the increase in the tree CD in each band. In a previous study, the results also found that the variance and variance ratio of images increased with the increase in CD values using IKONOS images [2,3]. Furthermore, the stability of spatial heterogeneity was also related to the sizes of the filter and the crowns. For the larger crowns, more pixels were covered in crowns, and the spatial heterogeneity of these pixels became more significant. It is confirmed that features of spatial heterogeneity with high sensitivity had a stronger ability to map tree CD values. However, it is not reliable to express the relationships between spatial heterogeneity and crowns in young forest ecosystems, because of just a few pixels in the crown.

In addition, texture features based on gray-scale co-occurrence matrices are widely used to express the spatial information of images [32,42,48]. Previous studies have shown that texture features of optical images with a high spatial resolution could distinguish forest crowns and gaps, which can help to diagnose the complexity of forest ecosystems [42,54]. In this study, eight common texture features were also obtained to map CD values. Compared to the results of spatial heterogeneity, the contributions of texture features in mapping tree CD were very limited in planted coniferous forest ecosystems. The results confirm that spatial heterogeneity had a stronger ability to map tree CD values using meter-level optical images than common textural features.

### 4.2. Matching of Variable Types and Feature Extraction Methods

Recently, spectral features (SFs) and spatial features (TFs and SH) have been proven to correlate with forest parameters, and an accurate extraction of spectral and spatial features from images is an important part in mapping forest parameters [32]. Essentially, filter-based and object-based feature extraction methods mainly focus on obtaining more homogeneous information and attenuating the effect of anomalous pixels on forest parameter mapping. By the average of pixels in the window, the filter-based feature extraction method does not completely reject the anomalous pixels but retains the spatial information of the image. On the other hand, by aggregating homogeneous pixels from an irregular object, the object-based feature extraction method obtains more homogeneous pixels and rejects anomalous pixels. Therefore, variations in spatial information between pixels can better reflect the sensitivity of spatial features, and the filter-based feature extraction method is more suitable for obtaining textural features and spatial heterogeneity. Previous results also confirmed that spectral features are more suitable for obtaining features by object-based methods [21].

In this study, filter-based and object-based methods were used to extract SFs, TFs, and SH from GF-2 images. The results inferred that the sensitivity between the tree CD and spatial features (TFs and SH) extracted by the filter-based method was higher than that extracted by the object-based method (Figure 9), and the accuracy of CD estimation using filter-based spatial features (TFs and SH) was higher than that obtained by the object-based method (Table 3 and Figure 11). Furthermore, the filter window sizes were also related to the stability and rationality of spatial heterogeneity. The results demonstrate that spatial heterogeneity's ability to map tree CD values was significantly stronger than spectral features after obtaining stable features with appropriate filter window sizes (larger than $17 \times 17$). Moreover, optimal results (Table 3 and Figure 13) were obtained by combination sets between different feature extraction methods (SFs (object) and SH (filter)).

In contrast, by achieving more homogeneous object pixels, it is more appropriate to extract spectral features (SFs) using the object-based method, and the stability and sensitivity of spectral features can be increased by eliminating the effect of anomalous pixels.

Furthermore, by using combination sets between different feature extraction methods, the "salt and pepper" phenomenon significantly decreased. It can be confirmed that the filter-based feature extraction method is more suitable for spatial features and the object-based feature extraction method is more suitable for spectral features.

## 5. Conclusions

In this study, to map tree CD values using meter-level optical remote sensing images, the spatial heterogeneity of pixels in crown was proposed to express the diversity of inclination angles of leaves based on the radiative transfer model and the relationships between the spectral reflectance of pixels and the CD were interpreted using simulated optical images and GF-2 images. Then, filter-based and object-based methods were also employed to extract three types of variables (SFs, TFs, and SH), and the tree CD values were successfully mapped by four models using three single types of variables and combinations variables with different strategies. The results demonstrate that the spatial heterogeneity of spectral reflectance is highly correlated with the CD and has more potential to describe changes in the sizes of the forest crown based on simulated and GF-2 images. The results also confirmed that the accuracy of mapped tree CD values varied with filter window sizes and segmentation scales, especially for spatial heterogeneity and combined variable sets. The optimal results were obtained by combination sets between different feature extraction methods (SFs (object) and SH (filter)). However, it was not reliable to express the relationships between spatial heterogeneity and crowns in young forest ecosystems. In the future, matching between the CD and spatial resolutions of images and the overlaps of forest crowns will be further investigated.

**Author Contributions:** Conceptualization, Z.L., J.L. and H.L. (Hui Lin); methodology, Z.L., K.D. and J.L.; software, Z.L., H.L. (Hao Liu) and T.Z.; validation, X.X., Z.L. and J.L.; formal analysis, Z.L., P.Y., T.Z. and J.L.; investigation, H.L. (Hui Lin), J.L., Z.L., Z.Y. and P.Y.; resources, H.L. (Hui Lin) and J.L.; data processing, Z.L., X.X., P.Y. and H.L. (Hao Liu); original draft, H.L. (Hao Liu) and Z.L.; review and revision, Z.L., J.L. and H.L. (Hui Lin).; final editing: Z.L. and J.L.; visualization, Z.L., J.L. and H.L. (Hui Lin); supervision, H.L. (Hui Lin) and J.L.; project administration, Z.L., H.L. (Hui Lin), and J.L.; funding acquisition, H.L. (Hui Lin), J.L. and Z.L. All authors have read and agreed to the published version of the manuscript.

**Funding:** This research was funded by the National Natural Science Foundation of China (project number: 32171784), the Innovative and Construction special funds of Hunan Province (project number: 2020NK2051), and the postgraduate scientific research Innovative project of Hunan province (project number: CX20210854).

**Data Availability Statement:** The observed GSV data from the sample plots and the spatial distribution data of forest resources presented in this study are available on request from the corresponding author. Those data are not publicly available due to privacy and confidentiality reasons. The GF-2 images are available from China Centre for Resources Satellite Data and Application website at http://www.cresda.com/CN/ (accessed on 10 May 2020).

**Conflicts of Interest:** The authors declare no conflict of interest.

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
