# Peer review of "Interpretation and Mapping Tree Crown Diameter Using Spatial Heterogeneity in Relation to the Radiative Transfer Model Extracted from GF-2 Images in Planted Boreal Forest Ecosystems"

_remotesensing, doi:10.3390/rs15071806_

Round 1

Reviewer 1 Report

1. In Section 2.1, the expression "the main species of trees are Chinese pine and larch " is inaccurate. It should be "The main tree species of artificial forest in the study area are Chinese pine and larch".

2. The first column "Tree Species" in Table 1 is incorrectly marked.

3. In Section 2.4.2, the expressions of Figure 4(a), Figure 4(b) and (c) are incorrect. They should be Figure 2(a), Figure 2(b) and (c).

4. This study collected data from 37 larch plots and 42 Chinese pine plots. Although both Chinese pine and larch are conifers, their tree shape and crown shape are different, should different tree species be studied separately? Therefore, the differences caused by different tree species should be taken into account in the result analysis.

Reviewer 2 Report

The manuscript entitled “Interpretation and Mapping Forest Crown diameter using Spatial heterogeneity related with the Radiative transfer model extracted from GF-2 images in planted boreal forest” is scientifically correct. Thus it is relevant for the Journal Forests MDPI after minor revision.

The authors have investigated a very specific issue (verification of the relationships between spatial heterogeneity and forest crown diameter (CD) with the purpose to improve the forest CD mapping) which can be useful to many scientific researchers in application of remote sensing for forest surveys and monitoring.

The structure of the manuscript is well organized. The authors have processes a big amount of data. Meanwhile, I have some minor comments which would like to be noted by the authors.

-        Section 3 Result/ p.10 / line 4 in the text below fig.5

“After the window size larger than forest CD, the spatial heterogeneity remains constant.”

·        Could you check the grammar in the first part of the sentence?

-        In several places the authors use the combination “difference methods”.

·       Could the authors please check the translation?

-        Section 3.4 Results of mapped CD using GF-2 images/ p.16

“Figure 12. Scatterplots between predicted and ground measured forest CD using difference variable sets with filter-based and object-based methods, the blue dashed line is the fitted line, and the color of points is determined by the residuals between predicted and ground measured forest CD.”

·        Would the authors please present more information/ explanation about "the fitted line"?

-        Section 3.4 Results of mapped CD using GF-2 images/ p.17

“Figure 13. The maps of forest CD using combined variable sets, (a1) and (b1) are combined variable sets (SF+TF and SF+SH) with filter-based method, respectively; (c1) and (d1) are combined variable sets (SF+TF and SF+SH) with object-based method, respectively; (e1) and (f1) are combined variable sets (SF(Object) & TF(Filter) and SF(Object) & SH(Filter)) between object-based and object-based method, respectively; (a2), (b2), (c2), (d2), (e2) and (f2) are enlarged maps of corresponding local areas, respectively.”

·        Would the authors please clarify “between object-based and object-based method” and specify the “local areas”?

I recommend that in section “Conclusion” the authors should more emphasize the importance of their research.

I recommend Minor Revision.

Good luck,

Reviewer

Reviewer 3 Report

The tropic of this manuscript is interesting because it deals with interpretation and mapping tree crown diameter using spatial heterogeneity related with the radiative transfer model extracted from GF-2 images in plantations. I agree that crow characteristic is an important parameter for forests and the results of this study could provide some information for plantation forest management. Whole text is easy to read and understand. I review this manuscript and provide following comment for authors.

1.      I provide two points for Title. Usually, “crown diameter” is used to individual tree level; therefore, use tree crown diameter might be suitable for Title. If this study addresses more than one plantation, authors could consider as “in planted boreal forests”.

2.      Abstract is suitable for this paper. I only have a slight suggestion. Forest crown diameter might consider as tree crown diameter and using this conception format whole text.

3.      In interdiction chapter, I suggest using point by point to show the objection of this study because that pattern is easy followed by international audiences.

4.      Please add sample number to Table 1 and each abbreviation should give full name and then use abbreviation.

5.      Each equation is individual and should have note below. Some note should be improved, such as “Where…” should be improved as “where…”. Please see Equations (1), (4), (5) and (7).

6.      The results are abundant that could interpret the characteristic tree crown diameter. I have no special comment here.

7.      Discussion chapter could be improved and the extending use of this study might be added to this chapter.

8.      Conclusion is suitable and I have no comment here.

Over all, I found that this manuscript is interesting and I recommend it for publication in Remote Sens after revised.
